# Probiotics, Proline and Calcium Induced Protective Responses of *Triticum aestivum* under Drought Stress

**DOI:** 10.3390/plants12061301

**Published:** 2023-03-14

**Authors:** Rima Mockevičiūtė, Sigita Jurkonienė, Vaidevutis Šveikauskas, Mariam Zareyan, Elžbieta Jankovska-Bortkevič, Jurga Jankauskienė, Liudmyla Kozeko, Virgilija Gavelienė

**Affiliations:** 1Laboratory of Plant Physiology, Nature Research Centre, Akademijos Str. 2, 08412 Vilnius, Lithuania; 2Department of Cell Biology and Anatomy, M.G. Kholodny Institute of Botany of the National Academy of Sciences of Ukraine, Tereshchenkivska Str. 2, 01601 Kyiv, Ukraine

**Keywords:** *lea* genes, prolonged drought, stress-protecting compounds, water deficit, winter wheat

## Abstract

In order to increase plants tolerance to drought, the idea of treating them with stress-protecting compounds exogenously is being considered. In this study, we aimed to evaluate and compare the impact of exogenous calcium, proline, and plant probiotics on the response of winter wheat to drought stress. The research was carried out under controlled conditions, simulating a prolonged drought from 6 to 18 days. Seedlings were treated with ProbioHumus 2 µL g^−1^ for seed priming, 1 mL 100 mL^−1^ for seedling spraying, and proline 1 mM according to the scheme. 70 g m^−2^ CaCO_3_ was added to the soil. All tested compounds improved the prolonged drought tolerance of winter wheat. ProbioHumus, ProbioHumus + Ca had the greatest effect on maintaining the relative leaf water content (RWC) and in maintaining growth parameters close to those of irrigated plants. They delayed and reduced the stimulation of ethylene emission in drought-stressed leaves. Seedlings treated with ProbioHumus and ProbioHumus + Ca had a significantly lower degree of membrane damage induced by ROS. Molecular studies of drought-responsive genes revealed substantially lower expression of Ca and Probiotics + Ca treated plants vs. drought control. The results of this study showed that the use of probiotics in combination with Ca can activate defense reactions that can compensate for the adverse effects of drought stress.

## 1. Introduction

Drought and heat stress have become the most important factors limiting crop, growth, development and yields [1]. Understanding the impact of the drought on crop production and most importantly, formulating smart strategies to withstand the drought while respecting the rules of sustainable agriculture is the challenge for scientists [1,2]. When plants suffer from drought stress, significant changes in their morphology can be observed. Usually, drought stress affect plant size. This adverse effect of water scarcity on crop plants causes fresh and dry biomass losses. Recently, a variety of new strategies have been devised to improve plant performance under environmental stress. Crop tolerance might be improved through several methods, including vegetation cover, plant breeding, genetic engineering, more croplands, or farm mechanization [3,4]. However, most of these solutions are time-consuming or cost-intensive, and may even aggravate climate change and environmental problems further. Different physiological operations have been exerted to reduce the negative effects of drought stress. For example, some researchers used compounds such as plant growth regulators, amino acids, calcium ions, and other chemicals for the recovery of plant growth under drought stress [5,6,7]. Currently, we find data that plant probiotics can significantly improve the growth and yield of winter wheat grown under drought conditions [8,9,10]. They can considerably reduce the effect of drought on winter wheat by enhancing the physiological aspects of the plant: RWC, membrane stability, chlorophyll content, water potential, proline, and sugar [5]. Such probiotics have the potential to be a long-term and successful method for reducing abiotic stressors, and a viable alternative for protecting plants exposed to abiotic stresses in the current context of fast-developing climate change [11]. There is still a lack of knowledge on the role of microbial biostimulants in winter wheat growth. Lack of water in plant cells causes stress, which manifests in wilting (due to reduced water levels), followed by further impaired growth and development. Plants may acquire drought resistance by activating primary protection mechanisms. It has been shown that calcium increases cell wall cohesion and improves the level of cell hydration that makes the cells drought-tolerant [12,13]. Calcium (Ca) ions are known as the main signal transmitters in controlling plant development and response to environmental stress conditions [14,15]. We find evidence that the positive effect of the exogenous application of calcium in improving stress tolerance can be attributed not only to the regulation of water status but also to antioxidant systems activity, osmolytes accumulation, improvement of photosynthetic pigment content and inducing of antioxidant genes [16,17].

It is reported in the literature that osmoregulation in plants under low water potential relies on the synthesis and accumulation of osmoprotectants or osmolytes such as soluble proteins, sugars, quaternary ammonium compounds, and amino acids, like proline. [18]. Proline is probably the most widely used amino acid to prevent losses due to abiotic stress. There are several research studies that support the exogenous application of proline to improve drought tolerance in maize [19], in barley [20] and in wheat [21,22]. Exogenous application of proline can mitigate the injurious impacts of drought stress on cereals associated with physiological characteristics [20,22]. Thus, the exogenous application of proline may be an effective approach to reducing the adverse effects of water stress, however, the potential role of proline in improving resistance to prolonged drought in winter wheat has not been investigated.

The considerable response of plants to drought that occurs at both cellular and molecular level is the expression of drought-responsive genes such as late embryonic protein (LEA) genes producing various types of proteins and enzymes [23]. We hypothesise that substances which protect plants from drought stress affect the expression of the *lea* gene family.

In order to increase the plant’s resistance to drought, the idea to treat the plants with calcium salt, plant probiotics, and proline exogenously is considered. The main objectives of this work were to compare and assess the effect of probiotics, proline, and calcium on biomarker responses of winter wheat to drought stress.

## 2. Results

### 2.1. Effect of Stress-Protecting Compounds on RWC of Wheat Seedlings Exposed to Prolonged Drought

Differences in RWC of wheat leaves appear on drought day 12 (Figure 1). A reduction of 5–10% indicates that the plants experienced mild drought stress according to the standard of Hsiao [24]. After 18 days of drought stress, all treatments had positive effects on RWC leaves. Winter wheat plants exposed to ProbioHumus and ProbioHumus + Ca had the best retention of leaf water content, 82% and 88% (moderate stress), respectively, while the RWC of control (untreated) wheat leaves was 61% (high stress according to Hsiao standard). The leaf RWC of proline-treated wheat seedlings was close to the control.

### 2.2. Impact of Stress-Protecting Compounds on Morphometric Parameters of Wheat Seedlings Exposed to Prolonged Drought Stress 

Drought stress negatively affected morphometric shoot parameters and was dependent on drought duration. On the 18th day of drought, shoot length was lower by 12%, fresh weight decreased by 74%, and dry weight by 20% as compared with watered control (Table 1). Wheat shoot length, FW and DW were significantly higher when drought-exposed plants were sprayed to proline, probiotics and grown in soil with incorporated Ca (Table 1). 

### 2.3. Impact of Stress-Protecting Compounds on Accumulation of Photosynthetic Pigments of Wheat Seedlings Exposed to Prolonged Drought Stress 

Quantitative analysis of pigments in wheat leaves showed that after 12 days of drought, the concentration of chlorophyll a decreased to 0.45 mg g^−1^ FW, while the chlorophyll b to 0.09 mg g^−1^ FW, as compared with watered control 1.2 mg g^−1^ and 0.49 mg g^−1^ FW, respectively. The tested preparations showed a significant effect on the accumulation of chlorophylls in wheat leaves as compared to the drought control. The highest chlorophyll ratio was found in wheat exposed to drought. All used stress-protecting compounds reduced chlorophyll ratio (Table 2).

### 2.4. Impact of Stress-Protecting Compounds on Ethylene Emission of Wheat Seedlings Exposed to Prolonged Drought

Ethylene emission in winter wheat leaves increased rapidly at the beginning of the drought, reaching 233.98 nL g^−1^ FW h^−1^ on the 6th day and 149.95 nL g^−1^ FW h^−1^ in irrigated wheat leaves (Figure 2). The evaluation of the effect of stress-protecting compounds on the ethylene emission in winter wheat leaves showed that, during the 6 days of drought treatments, the proline-applied plants showed a lower intensity of the proline-treated ethylene emissions in drought-treated plants, similar to that of the irrigated control. As the drought continued, ethylene emission in the leaves of the drought control decreased to 89.19 nL g^−1^ FW h^−1^ and 80.63 nL g^−1^ FW h^−1^ on days 12 and 18, respectively. On the 12th day of drought, ethylene emission significantly increased in Ca, proline, proline + Ca, treated plants. In contrast, the changes in ethylene content in the leaves of ProbioHumus and ProbioHumus + Ca treated plants did not change significantly from day 6 to day 18 of the drought (Figure 2).

### 2.5. Effect of Stress-Protecting Compounds on Biochemical Responses of Wheat Plant Exposed to Prolonged Drought

#### 2.5.1. Free Proline

The amount of free endogenous proline in plants exposed to 12 days of drought stress was 10.58 µmol g^−1^ FW and after 18 days of drought increased to 35.61 µmol g^−1^ FW, wheal in control plants it was 2.23–2.74 µmol g^−1^ FW (Figure 3). After 12 days of drought, the highest amount of free proline was detected in the variant sprayed with proline 34.02 µmol g^−1^ FW. As the drought continued for 18 days, the highest concentration of proline was detected in ProbioHumus and ProbioHumus + Ca treated plants, 1117.5 and 1194.1 µmol g^−1^ FW, respectively. Analyzing the results of the study of the free proline content, it was observed that the proline content significantly increases as the drought period continues.

#### 2.5.2. Hydrogen Peroxide (H_2_O_2_)

The H_2_O_2_ study showed that the levels of H_2_O_2_ in all drought-affected plants increased with the prolonged-drought duration (Figure 4). On the 12th day of drought, slight increases in H_2_O_2_ content were found in all test variants treated with stress-protective compounds. ProbioHumus + Ca was the most active drought stress protector, with a H_2_O_2_ content of 12.2 µmol g^−1^ FW after 18 days of drought, compared to 18.91 µmol g^−1^ FW for the control.

#### 2.5.3. Malondialdehyde (MDA)

Lipid peroxidation by MDA level showed that drought stress significantly increased the MDA content in winter wheat leaves: after 12 days of drought stress, the amount of MDA in the plants increased to 23.11 µmol g^−1^ FW, and after 18 days even up to 58.51 µmol g^−1^ FW, while in irrigated plants −16.05 and 15.35 µmol g^−1^ FW, respectively (Figure 5). In plants treated with ProbioHumus + Ca a significantly lower amount of MDA was detected on the 12th and 18 the day of drought −16.87 µmol g^−1^ FW and 27.64 µmol g^−1^ FW, respectively as compared with drought control. All other studied compounds reduced the amount of MDA in winter wheat leaves less effectively.

#### 2.5.4. PM ATPase Activity

Drought significantly reduced winter wheat leaves’ PM ATPase activity. All studied stress-protecting compounds increased the PM ATPase activity of drought-exposed plants on average from 63% to 308% compared to untreated plants. The highest enzyme activity was recorded in ProbioHumus + Ca treated plants on the 12th and 18th day of drought: 0.1445 and 0.231 μmol Pi mg^−1^ protein h^−1^ respectively. Proline treatment stimulated plant PM ATPase activity by 172 and 260% on day 12 and 18 of drought, respectively as compared with control (Figure 6).

### 2.6. Effect of Stress-Protecting Compounds on Late Embryogenesis Abundant (lea) Genes Expression Levels of Wheat Plant Exposed to Prolonged Drought

Quantitative results of RT-PCR analysis showed that among the 3 genes studied drought stress was most expressed in the *td27e* gene. The levels of *td27e* in the samples of plants treated with ProbioHumus + Ca are the lowest as compared with all tested variants (Figure 7). *td29b* gene expression levels were lower in Ca variants used both alone and together with proline and especially in combination with ProbioHumus and this was especially evident on the 18th day of the drought. The lowest levels of expression of *td11* gene were recorded in the samples of plants exposed to ProbioHumus + Ca at both tested drought durations (Figure 7).

## 3. Discussion

Wheat is a staple food for more than 35% of the world’s population, and its yield is significantly influenced by water resources scarcity [25]. Drought is a complex physicochemical phenomenon that affects morphological, physiological, biochemical and molecular processes in plants, resulting in growth inhibition [26,27]. Cell turgor, photosynthetic activity, oxidative metabolism, membrane stability are altered in response to water deficit and play an indispensable role in stress reduction. [28]. 

Scientists have been trying to overcome the impacts of drought by employing different strategies, such as foliar application of plant growth regulators, osmoprotectants, organic and inorganic nutrients which are efficient, economical and environmentally sound approaches [29,30]. According to literature, proline, plant probiotics, calcium have a positive effect on the processes of crop adaptation to abiotic stresses, including drought stress [10,17,30,31]. In the current study, the drought-stress compounds proline, Ca, ProbioHumus, and their combinations were tested. We hypothesized that the exogenous use of physiologically active substances can enhance plant growth and resistance to drought by maintaining water status, photosynthetic activity, osmoprotectant accumulation, and plasma membrane stability.

RWC in plants is a reliable measure of plant water status closely related to plant physiological function and shows the ability of the plant to retain water under drought conditions [27,30,32]. Indeed, wheat plants exposed to prolonged drought and treated with the studied compounds had positive effects on RWC of leaves. ProbioHumus and ProbioHumus + Ca had the most pronounced effect on water retention in leaves: RWC of treated plants after 18 days of drought was by 40% and 44% higher than in drought control. Water deficit negatively affected morphometric shoot parameters and was dependent on prolonged drought duration. Winter wheat growth and development: shoot length, FW and DW were significantly higher when drought-exposed plants were treated to proline, probiotics and grown in soil with incorporated Ca.

A number of studies suggest that leaf photosynthetic pigments content is the significant parameter to assess the severity of drought stress on various plants including wheat [20,33,34]. Decreasing chlorophyll levels is considered as a symptom of drought stress, which can be caused by a decrease in water content and damage to chloroplast membranes [27]. Our results showed that the decrease of total chlorophyll was slow during the first 6 days of water deficit (reduced to 6%), but decreased rapidly during the next six days to 68.7%. We found data about the positive effect of exogenous calcium [30,35] and plant microbial probiotics application on the chlorophyll content of wheat seedlings exposed to drought. It was reported that *Burkholderia phytofirmans*, *Bacillus megaterium* and *Bacillus licheniformis*, *Pseudomonas aeruginosa* significantly improve chlorophyll content under drought stress conditions [9,36,37]. Our study showed that the application of microbial probiotic preparation ProbioHumus and Ca increased the total content of chlorophyll under drought conditions. The effect of ProbioHumus + Ca on the amount of chlorophyll a and b was greater as compared to the application of probiotics or Ca alone. Researchers believe that this increase in photosynthetic pigments in plants treated with plant probiotics may be due to the activation of chlorophyll biosynthesis and limitations in the oxidative stress products, including H_2_O_2_ [37,38]. 

One of the earliest plant responses to adverse environmental factors is oxidative events in cells, which produce reactive oxygen species (ROS) such as H_2_O_2_ [39,40]. The maintenance of H_2_O_2_ levels in cells is particularly important due to its role in the establishment of cellular signaling cascades and stress-induced damage in plant responses to drought [32,41,42]. Our data on winter wheat showed that drought stress significantly increased H_2_O_2_ levels by a factor of 1.4 after 12 days of drought and by a factor of 2.9 after 18 days, compared to plants watered at the same age. This is in line with data found in the literature [30,40,42]. It is noted that at the onset of drought stress, elevated H_2_O_2_ levels have a protective effect by altering plant adaptation systems, but further increases can be toxic to cells, so maintaining balance is critical [42,43]. In the current study, the use of exogenous Ca, proline, ProbioHumus and their combinations had a positive effect on the maintenance of H_2_O_2_ homeostasis, especially when the drought was prolonged. The level of H_2_O_2_ in plants treated with proline + Ca, ProbioHumus + Ca was lower than in drought control. Abdelaal and colleagues [20] showed similar results with proline on barley. These results are in agreement with those obtained by Lalarukh et al. [9], who showed that winter wheat inoculated with *Pseudomonas aeruginosa* and Azeem et al. [37], who showed that maize plants treated with *Bacillus* strains reduced the formation of H_2_O_2_ under drought stress in comparison to uninoculated control plants. Our results showed that the addition of Ca to the soil has a positive effect on protecting winter wheat cells from oxidative damage. The concentration of H_2_O_2_ in the leaves of plants grown with Ca in all combinations was lower than in drought control plants. These data are complementary with the work of Khushboo et al. [30] in which the amount of H_2_O_2_ significantly decreased in drought stressed winter wheats treated with exogenous calcium. 

So, prolonged drought stress induces rapid and excessive production of H_2_O_2_ leading to lipid peroxidation and, consequently, membrane damage, enzyme inactivation, and protein degradation [2,30]. The amount of MDA a product of fatty acid peroxidation, is a suitable indicator of oxidative stress-induced membrane damage, which could reflect the degree of damage under adverse conditions [22,25,30,32]. Our results showed that when the plants were under mild drought stress on the 12th day, a 30% higher MDA content was recorded in the leaves of the drought-affected wheat compared to the well-watered plants. At this stage, only plants grown in soil with added Ca and treated with ProbioHumus had a significantly lower degree of membrane damage compared to drought control and other variants. At the stage of prolonged drought, all stress-protecting compounds decreased the damage caused by drought. These protective effects are consistent with studies in drought-stressed wheat [22,30] and barley [20]. The positive effect of ProbioHumus on increasing the antioxidant capacity of drought-affected wheat should be emphasized. The amount of MDA detected in probiotic-treated plant tissues was 39.4%, in Ca + probiotic-treated it was even 52.8% lower than that in drought control. Similar data on plant probiotics were obtained in the works of other researchers of winter wheat and corn plants [9,36,37,44]. 

It is now well established that PM ATPase activity changes depending on environmental factors [45,46,47], but conflicting data are obtained under drought stress. Some studies show that PM ATPase activity is enhanced [45,46,48,49], and other reports indicate inhibition of activity [47,50]. Our results show that the activity of PM ATPase decreased due to drought stress. Exogenous application of Ca, proline, and probiotics individually and in combination significantly increased PM ATPase activity of drought stressed plants vs. drought control. In the literature, we did not find any data on how probiotics affect ATPase activity during abiotic stresses. In our study, plants treated with ProbioHumus significantly increased PM ATPase activity under prolonged drought conditions. The highest ATPase activity was detected in ProbioHumus + Ca treated plants. When considering the effects of exogenous proline on PM ATPases under abiotic stress conditions, studies by Butt et al. [51] can be highlighted, which show that proline maintains ionic homeostasis by increasing ATPase activity. In our study, proline treatment significantly stimulated PM ATPase activity, but to a lesser extent than ProbioHumus + Ca. 

Proline is one of the best-known osmoprotectants, the accumulation of which is a physiological response of plants under drought stress that increases the ability of plants to survive under stress conditions [31,52,53]. Drought stress-induced increase in proline content in winter wheat was shown by several authors [10,30,40,54,55,56]. Our results agreed with the results of the mentioned studies. After 12 days of drought stress, plants accumulated 5-fold higher amounts of proline than the irrigated plants, as the drought continues after 18 days, the concentration of endogenous proline in plant leaves increases to 13-fold. Literature data showed that exogenously applied proline, probiotics, and other biostimulants increased endogenous proline and improved drought tolerance in plants [10,22,36,57]. The current study confirmed this opinion: after a prolonged drought, plants sprayed with exogenous proline had a 3.3-fold higher endogenous proline content than untreated plants. And also, plants exposed to the probiotic ProbioHumus produced a particularly high amount of endogenous proline both when treated alone and in combination with Ca. A greater increase in proline content in winter wheat plants supplemented with calcium compared to plants under drought stress alone was shown in Nayyar [55], and Khushboo et al. [30]. It is known that proline can accumulate to high concentrations during stress without damaging cellular macromolecules. Importantly, proline may help to stabilize proteins and protect cell structures from membrane damage, particularly when the stress becomes severe or lasts longer periods [2,58]. An explanation of how probiotics can promote proline accumulation during stress can be found in studies by other researchers [59,60,61]. They discuss that probiotics secrete osmolytes that act synergistically with plant-synthesized osmolytes, and some bacteria stimulate the production of osmoprotectants in host plants, thereby increasing plant drought resistance [62].

A number of reports have suggested that stress ethylene production in the plants increases dramatically during temperature and moisture extremes [63,64]. In our study, analysis of ethylene content shows a significant increase (2.08-fold) during the first 6 days of non-watering vs. watered plants. As the drought continued, on the 12th day, a sharp decrease in ethylene concentration was recorded. Meanwhile, in winter wheat plants exposed to Ca, proline and proline + Ca, a significant increase in ethylene content was recorded on the 12th day, followed by a significant decrease on the 18th day. Different situations of phytohormone ethylene emission were monitored in plants treated with probiotics alone and in complex with Ca during drought. In these variants, the amount of ethylene remained almost unchanged from the 6th day to the 18th day of drought. It can be that some rhizosphere bacteria are able to produce an enzyme that degrades the direct ethylene precursor and reduce the ethylene amount in the plant to resist root drying [65,66,67,68]. Our data contribute to the suggestion that inoculation of probiotic microorganisms affecting ethylene content may help to remove the inhibitory effect of drought stress on plant growth [2].

Recent advances have been made in determining how plants respond to drought by altering gene expression [23,69]. So far in wheat, there are identified several genes that are responsible for drought stress tolerance. These genes encode Late Embryogenesis Abundant (LEA) proteins, which help other proteins retrieve after denaturation during drought stress and play a water-binding role, helping to maintain minimal cellular water demand and protect plants from desiccation [69]. Our results confirmed *lea* genes participation in drought stress response. We detected, more transcripts of *lea* genes on the 12th day of drought vs. watered plants, Plus, the expression level of the *td27e* and *td29b* genes (belonging to *lea* family) in control wheat plants increased with increasing drought. Meanwhile, the expression level of the td11 gene decreased as the drought continued. It is thought that as the proteins encoded by *td29b* and *td27e* become more abundant, they may subsequently reduce *td11* expression [70]. In the current study, the expression of the studied *lea* genes was affected by Ca, Proline and ProbioHumus treatment during prolonged drought. After 12 days of drought, the expression level of all studied genes was significantly lower in stressed plants treated with probiotics + Ca vs. drought control. After 18 days of drought, the expression of *td27e* and *td29b* genes was significantly lower in variants with Ca, probiotics + Ca and proline + Ca vs. drought control. Gene expression of *td11* decreased in the variant with probiotics + Ca during prolonged drought.

The obtained results may highlight the mode of action of exogenous stress protective compounds in winter wheat. The ProbioHumus, proline and calcium treatments could help plants to invert the adverse effects of drought stress, and might play a key role in providing tolerance in plants through modulating RWC, ethylene, proline, total chlorophyll, ROS content, and PM ATPase activity and *lea* gene expression in winter wheat, thereby increasing plant growth. The data of the current study on winter wheat revealed that application of stress protecting compounds may activate a defensive response, which may compensate the negative impact of drought stress.

## 4. Materials and Methods

### 4.1. Plant Material and Growth Conditions

Wheat seeds (*Triticum aestivum* L. cv. ‘Skagen’) were sown in plastic cubic pots (15 × 35 cm) (21 pots in total), 50 seeds per pot, in a peat moss substrate (pH 5.5–6.5). Plants were germinated and grown under controlled conditions of a constant temperature of 23 ± 1 °C, a photoperiod of 16/8 h and a fluorescent light photon flux of 60 µmol m^−2^ s^−1^ at soil level. Soil moisture was maintained at ~60%.

#### Treatments

Drought treatment: during the drought, irrigation was stopped and the soil gradually dried out. Soil moisture was assessed using a soil moisture meter (Biogrod, China).

The following compounds were used for the drought stress control studies:(a)Calcium was added to the soil in the form of CaCO_3_ (MKDS) (hereafter referred to as Ca) at a rate of 3.71 g per pot, based on 70 g m^−2^.(b)ProbioHumus—probiotic concentrate (Baltic Probiotics, Rucavas pagasts, Latvia) was used for seed priming 2 µL/g and diluted with water 1:100 was used for seedling spraying 10 mL per pot at the 3–4 leaf stage (BBCH-scale 3–4) [71]. Composition of microorganisms: *Bacillus subtilis* (10^3^ CFU/mL), *Saccharomyces cerevisiae*, *Bifidobacterium animalis*, *B. bifidum*, *B. longum* (10^4^ CFU/mL), *Lactobacillus diacetylactis*, *L. casei*, *L. delbrueckii*, *L. plantarum*, (10^5^ CFU/mL), *Lactococcus lactis* (10^2^ CFU/mL), *Streptococcus thermophilus*, *Rhodopseudomonas palustris* and *R. sphaeroides* (10^4^ CFU/mL).(c)L-proline 1 mM aqueous solution (Roth) was used for seedling spraying 10 mL per pot at the 3–4 leaf stage (BBCH-scale 3–4) [71].

### 4.2. Experimental Design

Twenty-one pot was used for the experiment: three for rational watering and eighteen for drought simulation. Stress-protecting compounds were added according to the scheme: (1) watered control, (2) drought control, (3) Ca plus drought, (4) proline plus drought, (5) proline plus Ca and drought, (6) ProbioHumus and drought, (7) ProbioHumus plus Ca and drought (Figure 8).

### 4.3. Sampling

Plant samples were taken for analysis on three occasions: on the 6th day of the drought (soil moisture 40%), on the 12th day of the drought (soil moisture 20%), and on the 18th day when soil moisture was 12%. The watered plants, used as controls, were sampled at the same time (soil moisture 60%). Shoots of thirty wheat seedlings were sampled for morphometrical measurements. For biochemical and molecular analysis, three independent replicates were carried out using the third leaves of wheat plants. Freshly harvested samples were used for ethylene emission analysis and pigment measurement. For PM ATPase activity, MDA, H_2_O_2_, proline assays and RNA isolation, the samples were immediately frozen in liquid nitrogen and stored in a low-temperature freezer (Skadi Green line, International Labmate Ltd., St Albans, UK) at −80 °C until the analysis.

### 4.4. Morphometrical Measurements

Shoot length, fresh and dry mass were taken after 6, 12, and 18 days of growth using a ruler and balances (Kern EWJ).

### 4.5. Relative Water Content (RWC)

RWC was determined according to Weng et al. [32]. Fresh wheat germ leaves were collected and weighted as fresh weight (W_f_). After that, the leaves were left in the water for 24 h and weighted again after it to obtain a saturated weight (W_t_). The dry weight (W_d_) was obtained by drying the leaves in a drying chamber and weighted. RWC have been calculated according to the formula [32]:RWC = [(W_f_ − W_d_)/(W_t_ − W_d_)] × 100%(1)

### 4.6. Assessment of Biochemical Parameters

#### 4.6.1. Lipid Peroxidation According MDA

For analysis of MDA and H_2_O_2_, leaf material (0.5 g) was homogenized using 5% trichloracetic acid (TCA) (Sigma-Aldrich, St. Louis, MO, USA). The method of Hodges et al. [72] with slight modifications, have been used to estimate MDA. The homogenates were centrifuged at 12.13× *g* for 17 min (centrifuge MPW-351 R), and supernatant was added to 20% TCA containing 0.50% thiobarbituric acid (TBA) (Alfa Aesar, Haverhill, MA, USA). The homogenate was incubated in a heater at 95 °C for 30 min (Blockthermostat BT 200) and then subsequently cooled on the ice. The optical density was measured at 532 and 660 nm by spectrophotometer (Analytik Jena Specord 210 Plus, Analytik Jena, Jena, Germany). The results were expressed in µmol g^−1^ FW [72]. To avoid desiccation effects, the RWC was used as a factor to calculate the MDA content and to estimate the effective differences between treatments.

#### 4.6.2. H_2_O_2_

H_2_O_2_ content in leaves was determined according to Velikova et al. [73]. The supernatant was mixed with 10 mM, pH 7.0 potassium phosphate buffer (Alfa Aesar) and 1 M potassium iodide (Alfa Aesar) in a ratio of 1:1:2. The reaction solution was incubated for 30 min at 25 ° C in the dark. The absorbance of the supernatant was measured at 390 nm. The amount of H_2_O_2_ was calculated using a standard curve. The results expressed in µmol g^−1^ FW. 

#### 4.6.3. Proline

A slightly modified colorimetric reaction of Bates et al. [74] was used to determine the free proline content The equal volume of supernatant of a ground plant material (0.5 g), acetic acid, and acidified ninhydrin was mixed and heated for 1 h 15 min at 108.5 °C in a heater. The formed chromophore was extracted with toluene. The absorbance was read spectrophotometrically at 520 nm using a multi-sample quartz cuvette and Rainbow microplate reader, The corresponding content of proline was determined using the standard curve. Calculations were provided using the SLT program (SLT Labinstruments). Results were expressed as µmol of proline µmol g^−1^ FW.

#### 4.6.4. Photosynthetic Pigments

The photosynthetic pigments were extracted from fresh leaves with N,N′-dimethylformamide (DMF) (Sigma-Aldrich). Light absorption was measured at 480, 664, 647 nm. Chlorophyll a/b ratio and chlorophyll a and b contents were calculated according to Wellburn [75].

#### 4.6.5. Ethylene

Ethylene emission from freshly harvested leaf blades was evaluated according to Child et al. [76] with modifications. Samples with known mass were placed in 40 mL clear glass vials (Agilent technologies, Santa Clara, CA, USA) sealed with PTFE/Si septa caps and incubated for 24 h at 21 °C in darkness. Following incubation, 1 mL of gas sample from each vial was sampled using a gas-tight syringe (Agilent technologies) and injected into a gas chromatograph equipped with a stainless-steel column (Propac R, Sigma-Aldrich, USA) and hydrogen flame ionization detector. The temperature of the injector, column and detector was 110.90 and 150 °C, respectively. Helium (AGA) was used as the carrier gas. Calibrations were made with ethylene standard (Messer, Bad Soden, Germany). Results were expressed as nL g^−1^ FW h^−1^.

#### 4.6.6. PM ATPase Activity Assay

Membrane-enriched microsomal fraction was extracted from plant samples. Protein content was measured using the Bradford dye-binding procedure [77] at 595 nm. The H^+^-ATPase activity of microsomal fraction was evaluated according to released inorganic phosphate (P_i_) that accumulates as a result of ATP hydrolysis [78]. The colour reaction for P_i_ measurement was performed with ammonium molybdate and stannous chloride at 750 nm. The activity of PM ATPase was expressed as μmol P_i_ mg^−1^ of protein h^−1^.

### 4.7. Molecular Techniques

#### 4.7.1. RNA Extraction and Reverse Transcription

Total RNA was extracted from 200 mg of plant leaf material using the PureLink RNA Mini Kit (Ambion, Waltham, MA, USA) following the manufacturer’s recommendations and using Heraeus Fresco 21 Centrifuge (Thermo Scientific, Waltham, MA, USA). In order to avoid the contamination with genomic DNA, extracted total RNA was treated with RapidOut DNA Removal Kit (Thermo Scientific). The concentration and purity of treated RNA was evaluated with the spectrophotometer NanoPhotometer P330 (IMPLEN, Westlake Village, CA, USA). DNase-treated RNA samples were reverse-transcribed using the High-Capacity cDNA Reverse Transcription Kit (Applied Biosystems, Waltham, MA, USA) following the manufacturer’s recommendations. The obtained cDNA was stored at −20 °C.

#### 4.7.2. Real-Time Quantitative PCR

Real-time quantitative PCR was carried out using SYBR^®^ Green Universal Master Mix kit (Applied Biosystems) following the manufacturer’s recommendations and employing QuantStudio 3 real-time PCR system (Applied Biosystems). Two microliters of cDNA (the equivalent of 25 ng of total RNA) were used as a template for PCR. Cycling conditions comprised one cycle at 95 °C for 2 min and 40 cycles at 95 °C for 15 s following by 60 °C for 1 min. After the PCR run, a melting curve was generated and analyzed each time with the Quant Studio Design & Analysis Software v.1.5.2 (Applied Biosystems). Gene expression was calculated using 2^−ΔΔC^_T_ method [79].

#### 4.7.3. Primers

The sequences of primers used in the work were based on the sequences of LEA protein genes and were taken from the publication of Ali-Benali et al. [70]. The wheat mitochondrial 26S ribosomal RNA gene was used as a housekeeping gene [70]. The primer concentrations in all cases were 200 nM except for the *td27e*, where the concentration was 900 nM. The sequences of all primers used in the work are listed in the Table 3.

### 4.8. Statistical Analysis 

The results are presented as mean ± standard deviation (SD) of five independent experiments with at least three replicates. The data were analysed using one-way analysis of variance (one-way ANOVA). Tukey’s test was performed to test the statistical significance of differences (*p* <0.05) between means.

## 5. Conclusions

Exogenous application of stress-protecting compounds improved the prolonged drought tolerance of winter wheat. ProbioHumus and ProbioHumus + Ca had the greatest effect on maintaining the relative water content of the leaves and keeping plant growth parameters close to those of irrigated plants. The compounds tested delayed and reduced the stimulation of ethylene emission in drought stressed leaves. Plants exposed to the probiotic ProbioHumus produced particularly high levels of endogenous proline, both when exposed alone and in combination with Ca, helping to improve membrane integrity and maintain PM ATPase activity during long-term drought stress. Such plants had a significantly lower degree of membrane damage induced by ROS. Molecular studies of drought responsive genes revealed significantly lower expression in Ca, Ca + probiotics treated plants vs. drought control. The results of this study in winter wheat showed that the use of a probiotic composition in combination with Ca can activate defense reactions that can compensate for the negative effects of drought stress. It should be stressed that the compounds used in the study are suitable for organic farming, which underlines the relevance of this work.

## Figures and Tables

**Figure 1 plants-12-01301-f001:**
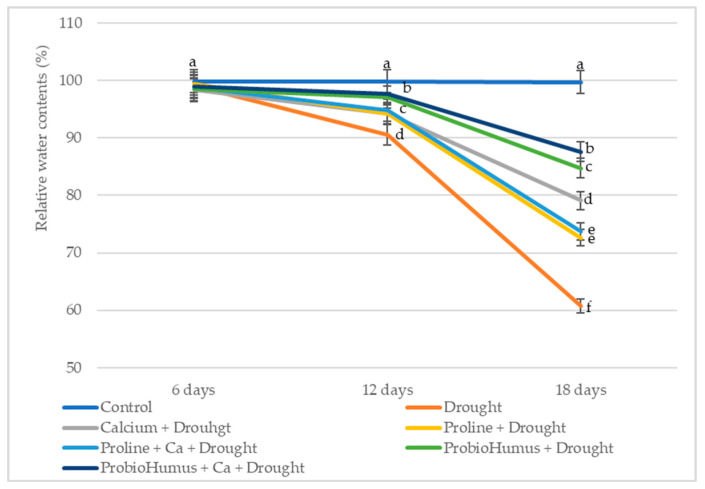
Impact of stress-protecting compounds on RWC of winter wheat leaves after drought stress. Values reported are mean of thirty plant leaves with standard deviation. Means with different letters in the same day of drought are significantly different (*p* < 0.05).

**Figure 2 plants-12-01301-f002:**
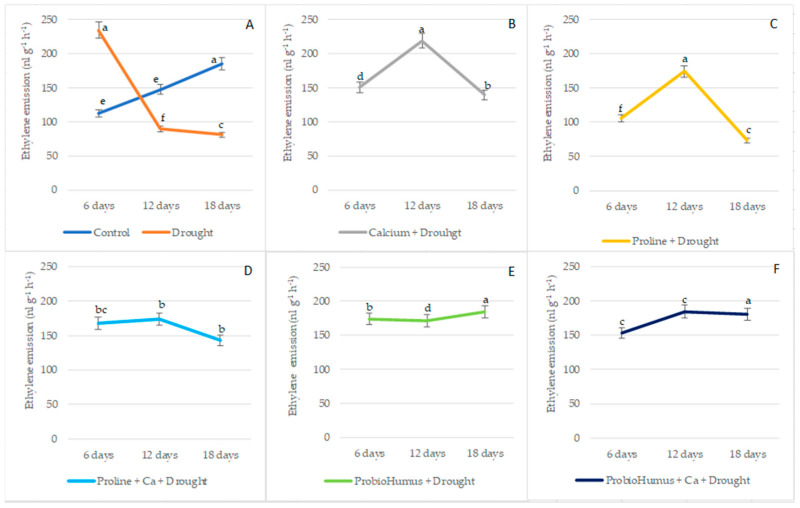
The effect of stress-protecting compounds application and prolonged drought stress (6, 12 and 18 days), on winter wheat ethylene emission level. Used types of application: (**A**)—water control and drought control; (**B**)—Ca + drought; (**C**)—proline + drought, (**D**)—proline + Ca + drought; (**E**)—ProbioHumus + drought; (**F**)—ProbioHumus + Ca + drought. Error bars represent the standard deviation of the mean. Different lowercase letters in the same day of drought indicate statistically significant difference (*p* < 0.05).

**Figure 3 plants-12-01301-f003:**
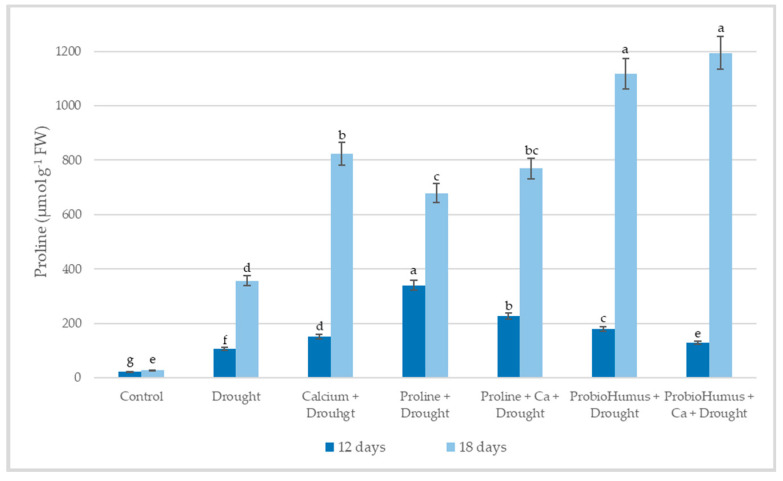
The effect of stress-protecting compounds application and prolonged drought stress (12, and 18 days), on winter wheat proline accumulation. Error bars represent the standard deviation of the mean. Control, Ca, proline, proline + Ca, ProbioHumus, and ProbioHumus + Ca indicates the type of application. Different lowercase letters in the same day of drought indicate statistically significant differences (*p* < 0.05).

**Figure 4 plants-12-01301-f004:**
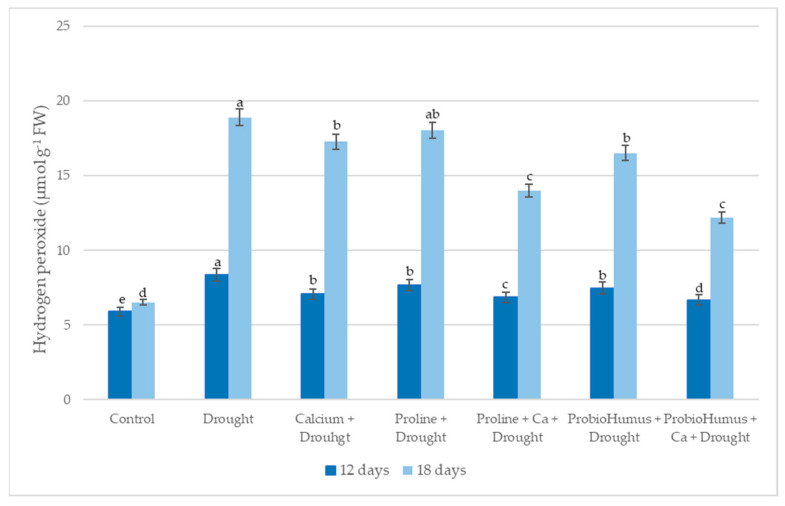
The effect of stress-protecting compounds application and prolonged drought stress (12 and 18 days), on winter wheat H_2_O_2_ content. Error bars represent the standard deviation of the mean. Control, Ca, proline, proline + Ca, ProbioHumus, and ProbioHumus + Ca indicates the type of application. Different lowercase letters in the same day of drought indicate statistically significant differences (*p* < 0.05).

**Figure 5 plants-12-01301-f005:**
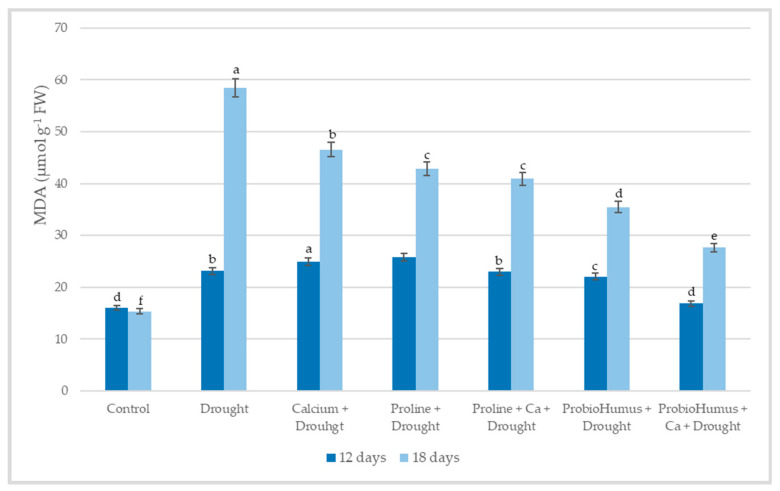
The effect of stress-protecting compounds application and prolonged drought stress (12 and 18 days) on winter wheat MDA content. Error bars represent the standard deviation of the mean. Control, Ca, proline, proline + Ca, ProbioHumus, and ProbioHumus + Ca indicates the type of application. Different lowercase letters in the same day of drought indicate statistically significant differences (*p* < 0.05).

**Figure 6 plants-12-01301-f006:**
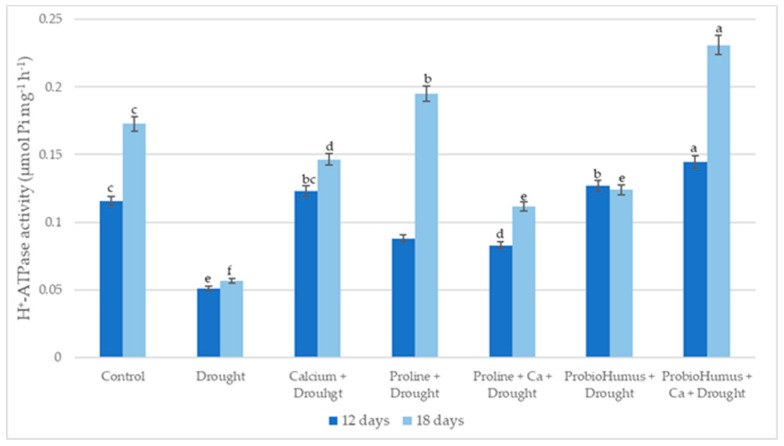
The effect of stress-protecting compounds application and prolonged drought stress (12 and 18 days) on winter wheat PM ATPase activity. Error bars represent the standard deviation of the mean. Control, Ca, proline, proline + Ca, ProbioHumus, and ProbioHumus + Ca indicates the type of application. Different lowercase letters in the same day of drought indicate statistically significant differences (*p* < 0.05).

**Figure 7 plants-12-01301-f007:**
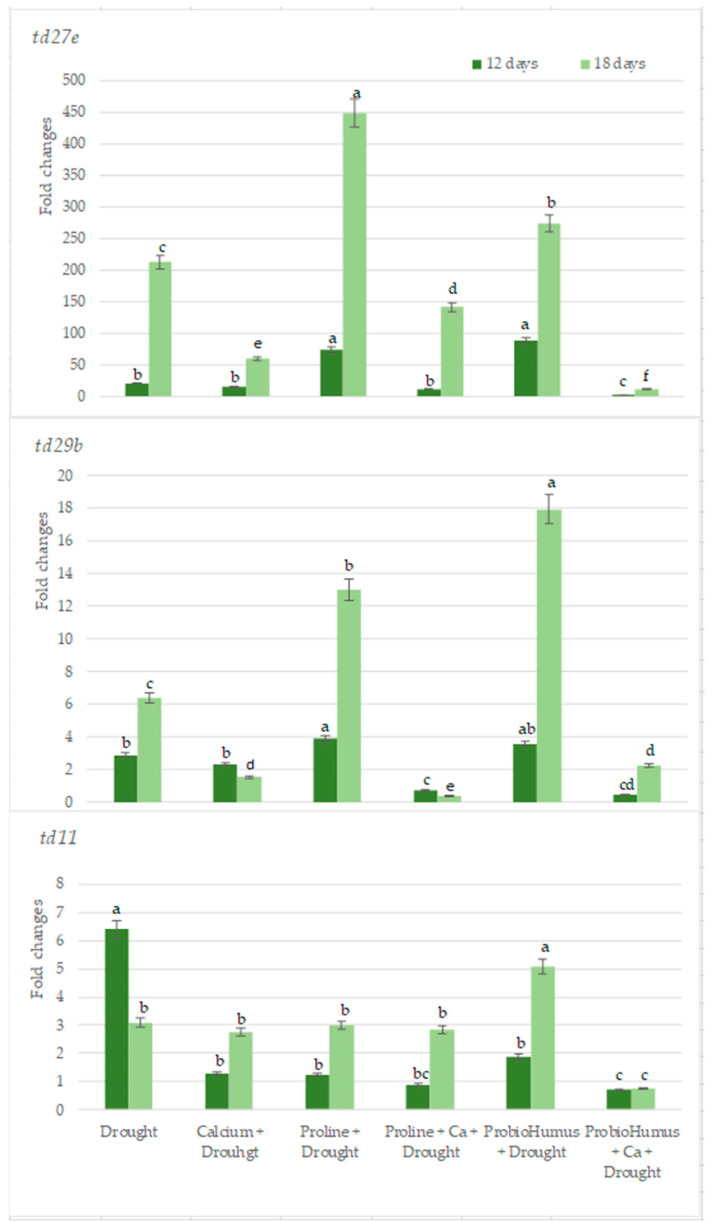
The effects of stress-protecting compounds application and prolonged drought stress (12 and 18 days) on winter wheat *lea* genes (*td27e*, *td29b*, *td11*) expression level. Error bars represent the standard deviation of the mean. Control, Ca, proline, proline + Ca, ProbioHumus, and ProbioHumus + Ca indicates the type of application. Different lowercase letters in the same day of drought indicate statistically significant differences (*p* < 0.05).

**Figure 8 plants-12-01301-f008:**
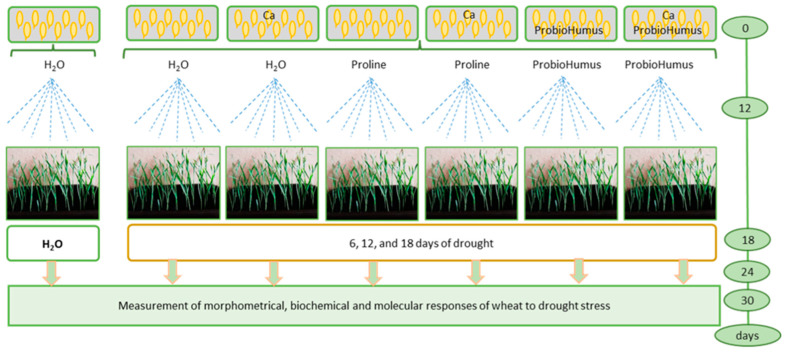
Experimental design.

**Table 1 plants-12-01301-t001:** Effect of stress-protecting compounds on morphometric parameters of winter wheat seedlings (per plant).

Treatment	Average Length (cm)			Average Weight (g)					
				Fresh			Dry		
	6 Days	12 Days	18 Days	6 Days	12 Days	18 Days	6 Days	12 Days	18 Days
Control	33.17 d	35.80 c	39.45 a	0.26 c	0.33 d	0.60 a	0.032 d	0.040 e	0.065 b
Drought	32.06 e	33.67 e	34.65 e	0.23 e	0.31 e	0.15 f	0.025 e	0.037 f	0.052 e
Calcium + Drouhgt	33.20 d	35.07 d	38.32 b	0.28 b	0.37 c	0.26 c	0.035 c	0.044 d	0.062 c
Proline + Drought	33.96 c	36.22 c	35.23 d	0.25 d	0.32 de	0.16 ef	0.032 d	0.044 d	0.060 d
Proline + Ca + Drought	33.98 c	36.76 bc	37.45 c	0.27 bc	0.39 b	0.16 e	0.039 b	0.051 b	0.063 c
ProbioHumus + Drought	34.28 b	37.12 b	38.43 b	0.27 bc	0.38 bc	0.20 d	0.040 b	0.049 c	0.065 b
ProbioHumus + Ca + Drought	35.85 a	39.38 a	39.74 a	0.31 a	0.42 a	0.28 b	0.049 a	0.055 a	0.070 a

Different letters in columns designate statistically significant difference at *p* < 0.05.

**Table 2 plants-12-01301-t002:** Effect of stress-protecting compounds on chlorophyll content of winter wheat seedlings.

Treatment	Chlorophyll a		Chlorophyll b		Chlorophyll Ratio a/b	
	mg g^−1^ FW					
	6 Days	12 Days	6 Days	12 Days	6 Days	12 Days
Control	1.27 b	1.19 a	0.53 a	0.54 a	2.38 bc	2.21 e
Drought	1.20 e	0.45 g	0.49 d	0.09 e	2.45 b	4.92 a
Calcium + Drouhgt	1.21 de	0.54 d	0.51 bc	0.14 cd	2.35 c	3.73 d
Proline + Drought	1.24 c	0.46 f	0.50 c	0.12 d	2.46 b	3.69 d
Proline + Ca + Drought	1.32 a	0.49 e	0.51 bc	0.13 cd	2.57 a	3.65 d
ProbioHumus + Drought	1.22 d	0.59 c	0.52 b	0.15 c	2.34 c	3.92 bc
ProbioHumus + Ca +Drought	1.27 b	0.75 b	0.53 a	0.17 b	2.39 bc	4.15 b

Different letters in columns designate statistically significant difference at *p* < 0.05.

**Table 3 plants-12-01301-t003:** The sequences of primers used in the work.

Gene	LEA Proteins Group	Primer Pairs	Primer Sequences (5′-3′)
26S		F/R	CCGGTTGTTATGCCAATAGCA/GCGGCGCAGCAGTTCT
Td11	2	F/R	AGGTGATCGATGACAACGGTG/ACCCTCGGTGTCCTTGTGG
Td29b	4	F/R	CGCACCCAGCTAGTAAGTTCG/CCCAGCCCAGTAATAACCCAT
Td27e	2	F/R	CAGCACTGAGCCGACGG/ACGTGGAACTAGAAGGCATTGAC

## Data Availability

The data supporting the reported results can be found in the archive of scientific reports of the Nature Research Centre.

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
