# Peer review of "Probiotics, Proline and Calcium Induced Protective Responses of Triticum aestivum under Drought Stress"

_plants, 2023, doi:10.3390/plants12061301_

Round 1
Reviewer 1 Report
Comments on Manuscript ID: plants-2105772 “Probiotics, Proline and Calcium Induced Protective Responses of Wheat under Drought Stress”
The authors consider the possibility of increasing the drought tolerance of plants by exogenously treating them with stress-protective compounds. The idea of using probiotics in complex with calcium as stress-protective compounds is quite new and promising. The studies were carried out under controlled conditions, simulating prolonged drought. Special value of the paper due to the fact that the authors have combined several methods of Plant Biology: physiological, biochemical and molecular genetics.
The article is written professionally, but I have a few comments:
1) the authors use the plant name "wheat" in the title, I suggest including the Latin name of the plant to avoid confusion between two cultivated species inside wheat genera, also English name of the species should have a couple of words, the same suggestion is valid for the titles of the tables and figures
2) there are two repeated references 32 and 72,
3) in L507 if you say plasma membrane ATPase, please do not duplicate and write PM ATPase,
4) you should not repeat the term hydrogen peroxide after you give the formula, you should only write the formula below,
5) I noticed that the authors did not mention about the use of compounds suitable for organic farming in there study, which could underline the relevance of the study.
Author Response
Dear Prof. Dr. Renata Szymańska and Dr. Aleksandra Orzechowska,
With this resubmission, please find our revised manuscript (plants-2105772) for consideration to be published in the Special Issue „Responses of Plants to Environmental Stresses Volume II”.
We are grateful for the valuable and constructive comments of the reviewers. We have carefully revised the manuscript and made some changes: we have provided point-by-point responses to the comments of reviewers. Hopefully, the subsequent modifications are satisfying for both reviewers.
Please find our responses to all the comments and recommendations of reviewers below. We look forward to Your positive response.
Sincerely,
Dr. Rima Mockevičiūtė
Nature Research Centre, Institute of Botany
Akademijos Str. 2, Vilnius LT-08412, Lithuania
Phone: +370 5 2729839
E-mail: rima.mockeviciute@gamtc.lt
Response to Reviewer 1 Comments
Point 1: the authors use the plant name "wheat" in the title, I suggest including the Latin name of the plant to avoid confusion between two cultivated species inside wheat genera, also English name of the species should have a couple of words, the same suggestion is valid for the titles of the tables and figures.
Response 1: Thank You for the comment. Following the recommendations, we have used the Latin name of the species Triticum aestivum instead of the word "wheat" in the title. The English plant name has also been replaced by a couple of words "winter wheat" throughout the text, as wheat cultivars are classified by growing season, such as winter wheat vs. spring wheat.
Point 2: there are two repeated references 32 and 72,
Response 2: Thank You for your comment. We changed cited literature sources.
Point 3: in L507 if you say plasma membrane ATPase, please do not duplicate and write PM ATPase,
Response 3: Thank you for your valuable comment. We have revised the text to avoid duplication and used the PM ATPase throughout the manuscript as recommended.
Point 4: you should not repeat the term hydrogen peroxide after you give the formula, you should only write the formula below,
Response 4: Thank you for your valuable comment. We edited the text according to your recommendations.
Point 5: I noticed that the authors did not mention about the use of compounds suitable for organic farming in there study, which could underline the relevance of the study.
Response 5: Thank you for your valuable comment. We added the text according to your recommendations.
Response to Reviewer 2 Comments
Point 1: Statistical analysis: (lines 496-497: “The data were analysed using analysis of variance (ANOVA).”: What model was used? These results must be previously presented to understand the weight of each main factor (rootstock and salt level) and the respective interaction. Regarding the experimental device, I expected that the stress-protecting compounds would also be applied to plants without water stress. Authors must justify the option followed.
Response 1: Thank you for your valuable comment. The data were analysed using one-way analysis of variance one-way ANOVA. We edited MM section according to Reviewer 2 recommendations. Many thanks to the reviewer for the idea of using stress-protecting compounds in irrigated plants, and we will be sure to do this research in the future.
Point 2: Some metabolites (chlorophylls, proline, etc.) were expressed by fresh weight. As plants have different RWC, the expression by dry weight is more adequate in order to nullify that effect and thus to evaluate the effective differences between treatments.
Response 2: Thank you for your comment. To avoid desiccation effects, the RWC was used as a factor to calculate the metabolite content per 1 g of FW and to estimate the effective differences between treatments. We added the explanation to the text in the MM section.
Point 3: In Table 2, in addition to the parameters presented, I suggest also the inclusion of the chlorophyll a/b ratio and the carotenoid content. Both parameters can help to better discuss the obtained results. Still in this table, the data obtained after 18 days of drought stress were not presented. Why? On the other hand, figures 3, 4, 5, 6 and 7 do not show the results obtained after 6 days... It will be better to justify the procedure followed.
Response 3: Thank you for your comment. We have added the chlorophyll a/b ratio to Table 2 and edited the results section as recommended. It was appropriate to look for changes in ethylene emission after 6 days at the very beginning of the drought stress when the RWC of the plant leaves was only reduced by 1-2% (Figure 1).
Point 4: Line 155-102160: Please, review this text. I cannot see these results in figure 4.
Response 4: Thank You for the valuable comment. We are very sorry for the mistake, we have changed Figure 4.
Point 5: Line 391 (Experimental design): How do you classify the water stress level of this experience? Moderate? Severe? In the description, it is not clear how this stress was imposed. Please the description needs to be improved. Soil moisture 40% or 20% - Evaluation by weight of pots?
Response 5: The level of drought stress was estimated according to the standard of Hsiao L86-87, L90-91. A description of how the drought stress was induced is given MM section. Soil moisture was assessed using a soil moisture meter (Biogrod, China).
Point 6: Keywords - they should be organized alphabetically.
Response 6: The list of keywords was edited according to the comment of Reviewer 2: lea genes; prolonged drought; stress-protecting compounds; water deficit; winter wheat.

Reviewer 2 Report
The topic is scientifically interesting and falls within the scope of this journal. The authors presented an important study whose main purpose was to evaluate the impact of exogenous calcium, proline, and plant probiotics on the response of winter wheat to drought stress. The objectives of this manuscript are clear and precise. The manuscript seems to be well-designed and written and could be very suitable to increase the knowledge about the best strategy to mitigate the negative effects of drought in winter cereal crops. My biggest criticism is related to methodological issues. I suggest that they be considered in order to improve the quality of the article. More accurately, I have the following specific comments:
1) Statistical analysis: (lines 496-497: “The data were analysed using analysis of variance (ANOVA).”: What model was used? These results must be previously presented to understand the weight of each main factor (rootstock and salt level) and the respective interaction. Regarding the experimental device, I expected that the stress-protecting compounds would also be applied to plants without water stress. Authors must justify the option followed.
2) Some metabolites (chlorophylls, proline, etc.) were expressed by fresh weight. As plants have different RWC,the expression by dry weight is more adequate in order to nullify that effect and thus to evaluate the effective differences between treatments.
3) In Table 2, in addition to the parameters presented, I suggest also the inclusion of the chlorophyll a/b ratio and the carotenoid content. Both parameters can help to better discuss the obtained results. Still in this table, the data obtained after 18 days of drought stress were not presented. Why? On the other hand, figures 3, 4, 5, 6 and 7 do not show the results obtained after 6 days... It will be better to justify the procedure followed.
4) Line 155-102160: Please, review this text. I cannot see these results in figure 4.
5) Line 391 (Experimental design): How do you classify the water stress level of this experience? Moderate? Severe? In the description, it is not clear how this stress was imposed. Please the description needs to be improved. Soil moisture 40% or 20% - Evaluation by weight of pots?
6) Keywords - they should be organized alphabetically.
Author Response

(The authors gave the same response as above.)

Round 2
Reviewer 2 Report
The responses given to points 1 and 2 are not convincing. A statistician will be able to give an opinion on the procedure followed. Regarding point 2, I completely disagree with the answer given by the authors. I repeat "As plants have different RWC, the expression by dry weight is more adequate in order to nullify that effect and thus to evaluate the effective differences between treatments."
Author Response
Dear Prof. Dr. Renata Szymańska and Dr. Aleksandra Orzechowska,
With this resubmission, please find our revised manuscript (plants-2105772) for consideration to be published in the Special Issue „Responses of Plants to Environmental Stresses Volume II”.
We are thankful for the valuable comments of the reviewer. We revised the manuscript according to the comments.
Please find our responses to the comments below. We look forward to Your positive response.
Sincerely,
Dr. Rima Mockevičiūtė
Nature Research Centre, Institute of Botany
Akademijos Str. 2, Vilnius LT-08412, Lithuania
Phone: +370 5 2729839
E-mail: rima.mockeviciute@gamtc.lt
Response to Reviewer 2 Comments
Point 1: The responses given to points 1 and 2 are not convincing. A statistician will be able to give an opinion on the procedure followed.
Response 1: Thank you for the comment. The data were analysed using a one-way analysis of variance (one-way ANOVA). We edited captions of tables and figures by adding text explaining the meaning of compared values.
Point 2: Regarding point 2, I completely disagree with the answer given by the authors. I repeat "As plants have different RWC, the expression by dry weight is more adequate in order to nullify that effect and thus to evaluate the effective differences between treatments."
Response 2: We apologise for not explaining the calculation clearly enough. Bearing in mind, that plants are wilting under drought we were looking for ways to calculate the metabolite content. After studying the scientific literature on plant drought stress we found that studies provide different ways of estimating metabolite concentrations in the plant (conversion to dry mass, or to fresh mass by estimation of RWC, etc.). Thus, in this article in order to nullify the drought effect (wilting), we took into account the RWC values as coefficients for estimating tested biochemical parameters.

Round 3
Reviewer 2 Report
I accept the justifications given by the authors and therefore I have nothing to object to their publication.